# Environmental health management in local community isolation facilities during COVID-19 pandemic: A case study in Nakhon Si Thammarat, Thailand

Arif Cheseng[1], Udomratana Vattanasit[1,2]*, Jira Kongpran[1,2], Chamnong Thanapop[1,3], Maisarah Nasution Waras[4]

1 Master of Public Health Program, School of Public Health, Walailak University, Nakhon Si Thammarat, Thailand, 2 Department of Environmental Health and Technology, School of Public Health, Walailak University, Nakhon Si Thammarat, Thailand, 3 Department of Community Public Health, School of Public Health, Walailak University, Nakhon Si Thammarat, Thailand, 4 Department of Toxicology, Advanced Medical and Dental Institute, Universiti Sains Malaysia, Kepala Batas, P. Pinang, Malaysia

* udomratana.va@mail.wu.ac.th

## Abstract

Community isolation facilities (CIFs) were adopted as a containment strategy in many countries during the COVID-19 pandemic, but information on environmental health management in CIFs is unknown. This retrospective research aimed to study the preparedness and implementation of CIFs from an environmental health perspective. The study was conducted in Nakhon Si Thammarat Province, Thailand. A total number of 114 staff working during the establishment and operation of 57 CIFs were enrolled. Two questionnaires were collected from the founding and operating staff from 27 May to 5 October 2022. One questionnaire, designed based on the guidelines for establishing of CIFs developed by the Department of Health, was used to examine the preparedness of the founding staff. Another questionnaire was developed to investigate the implementation by the operating staff. The data was presented as the levels of preparedness and implementation of CIFs and analyzed by descriptive statistics. Local government organizations played a major role in the establishment and cooperation with local public health offices in the operation of CIFs. Two-thirds of the founding and operating staff had no experience in environmental health. However, most of the CIFs showed preparedness for the establishment of CIFs and conducted environmental health implementation at a good level in all dimensions, except for infrastructure for wastewater treatment, disinfection, and system monitoring. The decentralized governance model in Thailand facilitated the operation of small CIFs to prevent COVID-19 transmission in local communities. However, qualified personnel and appropriate infrastructure were obstacles to full environmental health implementation. The guidelines by the Department of Health suggested as fundamental for environmental health management of CIFs in this

**Data availability statement:** Relevant data are within the manuscript and its Supporting Information files.

**Funding:** The author(s) received no specific funding for this work.

**Competing interests:** The authors have declared that no competing interests exist.

study should be appropriately adopted based on the different contexts of each country to ensure preparedness for future infectious disease outbreaks.

## Introduction

The coronavirus disease 2019 (COVID-19) is a disease caused by a respiratory infection of the novel coronavirus, the severe acute respiratory syndrome coronavirus 2 (SARS-CoV-2). Cases of COVID-19 first emerged in Wuhan, China in December 2019 [1]. In Thailand, the first case of COVID-19 reported on 13 January 2020, was a Wuhan resident who traveled to Bangkok [2]. This was the first COVID-19 case found outside China [3]. The first Thai case, a taxi driver infected with SARS-CoV-2 from Chinese tourists, was reported on 31 January 2020 [4]. On 11 March 2020, the World Health Organization (WHO) designated the outbreak as a pandemic [3].

During 2021−2022, Thailand faced the fourth and fifth waves of domestic COVID-19 infection [5]. The number of new infections was rapidly increasing to the point of overwhelming the capacity of public health services, i.e., hospitals, emergency field hospitals, and hospitels (hotels repurposed as temporary hospitals). The approach that the Royal Thai Government implemented to solve the problem was the classification of COVID-19-infected patients according to their symptoms. Accordingly, COVID-19 patients with mild or asymptomatic symptoms (Green Group) who did not require hospitalization and advanced medical equipment could be isolated. Therefore, the number of bed occupancy could be preserved for COVID-19 patients who have moderate to severe symptoms.

Thailand applied both home-based isolation (HI) and community-based isolation (CI) for the COVID-19 confirmed cases in the Green Group [6]. HI has been commonly adopted in European countries and the USA for infected patients with mild or asymptomatic symptoms of COVID-19 [7–9]. With this isolation approach, it was difficult to monitor for the disease progression. Also, HI was unlikely to be fully enforced since the patients might not strictly stay at home. Moreover, direct contact with the SARS-CoV-2 from COVID-19 patients or their household infectious waste could lead to transmission to their family members and the community. Compared with HI, CI was more widely applied in Asian countries, for example, China [10], South Korea [11,12], Japan [13], Singapore [14], Vietnam [15], and Thailand [6].

The operation of community isolation facilities (CIFs) provided fast diagnosis with routine basic medical screening for COVID-19 patients and communication with contract hospitals. Therefore, the progression and severity of the disease could be monitored, and the patients could be transferred to the hospital if necessary. Importantly, CI is an effective containment measure that minimizes the movement and contact of the patients, prevents further community transmissions, and accordingly suppresses the number of new infections in communities. In Thailand, CI was the primary isolation strategy during the Delta variant outbreak. CIFs were gradually phased out due to the decline in the situation and budget constraints. Then, HI became the primary option with the emergence of the Omicron variant, which led to rapid transmission and a surge in cases that were asymptomatic or had mild symptoms.

Thailand is the only upper-middle-income country in the top five of the Global Health Security (GHS) Index among 195 countries in 2021 [16]. This high ranking reflects the effectiveness of the government's public health policy and emergency management during the COVID-19 pandemic. Prompt response to patients with mild or asymptomatic symptoms of COVID-19 by adopting the CI strategy played a significant role in effective disease control in Thailand. The containment of COVID-19 patients in CIFs could suppress the spread of SARS-CoV-2 to the community by preventing transmission via direct contact with the patients and indirect contact through environmental pathways.

Up to the present, there have been a huge number of studies on COVID-19 published since the first emergence of the disease in 2019. The research was presented on various aspects of the COVID-19 pandemic, including environmental health concerns, such as biomedical waste management [17], municipal solid waste management [18], emerging contaminants in wastewater [19], linkages between environmental pollution and COVID-19 infection rates [20] or severity [21]. Until now, however, no research has comprehensively studied environmental health issues related to the COVID-19 pandemic. Holistic environmental health management would be beneficial for effectively reducing the spread of COVID-19 infections. Moreover, information on environmental health management in CIFs is unknown. Exploring this information is essential to ensure the success of CIFs' operation.

This retrospective research aimed to study the preparedness and environmental health implementation in local CIFs in Nakhon Si Thammarat Province, Thailand, during the COVID-19 pandemic. In October 2021, during the study's preparatory phase, the Ministry of Public Health designated 10 of Thailand's 77 provinces as a "watchlist," which included Tak, Ratchaburi, Chantaburi, Rayong, Nakhon Ratchasima, Nakhon Si Thammarat, Narathiwat, Pattani, Yala, and Songkhla [22]. Local Government Organizations (LGOs) were central to the establishment of CIFs. Nakhon Si Thammarat was purposively selected as the study site. Although it had the second-largest number of LGOs on the watchlist, surpassed by Nakhon Ratchasima [23], it was chosen for its geographical convenience to the researcher, which facilitated data collection amidst the pandemic's travel restrictions. This study marks the first to focus on environmental health management in CIFs. It was crucial to ensure appropriate environmental implementation to prevent the severity of the disease or complications of the patients from environmental factors and to protect people in the community from viral transmission. Lessons learned from this study on environmental health implementation in CIFs would be beneficial for infectious disease emergency response in the future.

## Methods

### Study locations and study subjects

The operation of CIFs in Thailand was initiated by the Centre for COVID-19 Situation Administration (CCSA) under the Emergency Decree on Public Administration in Emergency Situations, B.E. 2548. The Prime Minister directly commanded the CCSA, working with the Ministry of Public Health and related government agencies. The levels of organizations involved in establishing and operating the CIFs are demonstrated in Fig 1. The establishment of the CIFs in each province was managed under orders from the Permanent Secretary of the Ministry of Interior. The Provincial Governor put measures in place for the LGOs to act as the host of CIFs. In addition, local public health offices also acted as the host and cooperated with LGOs to operate the CIFs.

The study sites were CIFs established in Nakhon Si Thammarat Province, located in the South of Thailand. Two main eligibility criteria were used for selecting CIFs in this study. Firstly, the CIFs had to be already in operation. CIFs that were registered but had no occupied patients were excluded. Secondly, both the founding and operating staff were available to provide data during the study period. During the study period, there were 114 CIFs in the province. Out of these, 57 CIFs met the criteria and consented to participate in the study. Accordingly, 57 founding and 57 operating staff members were enrolled in the study. The descriptions of eligibility criteria to enroll the study subjects are presented as follows.

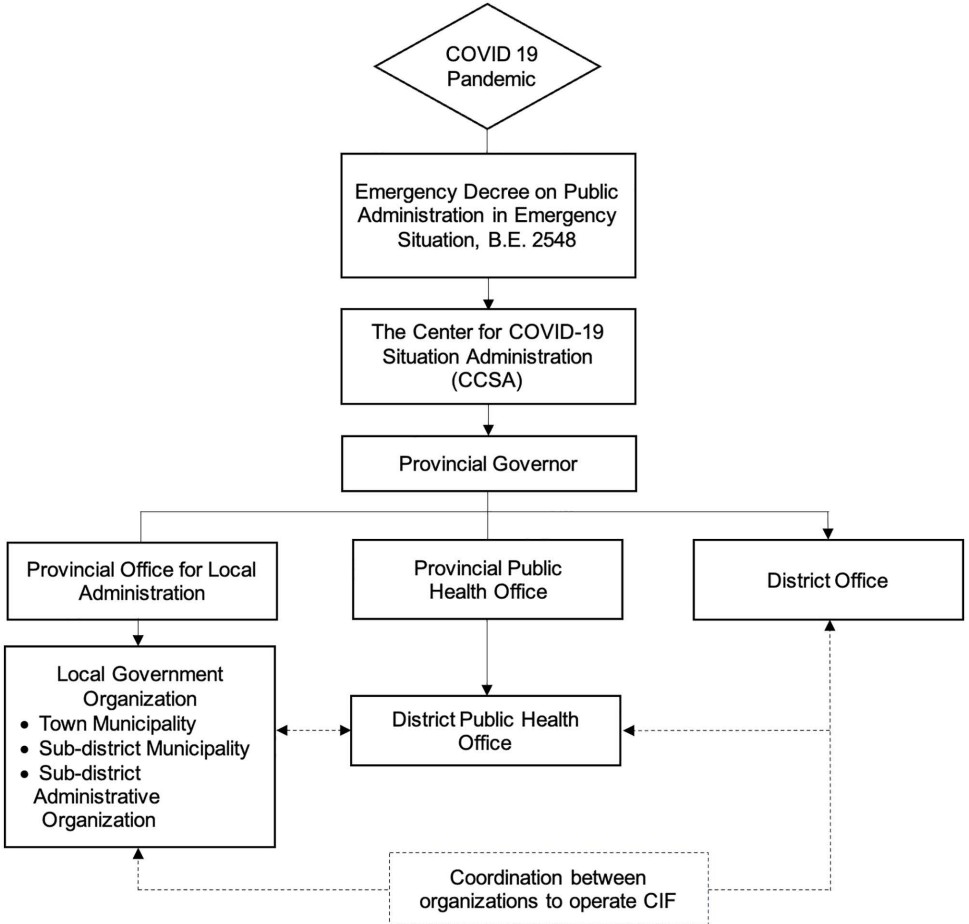

**Fig 1. Organizations involved in the establishment and operation of the CIFs.** The dotted line represents coordination between organizations to operate CIFs.

### Inclusion criteria

(1) The founding staff must have been a government officer or a private CIF owner who participated in establishing the CIF.

(2) The operating staff must have been a practitioner involved in implementing environmental health activities during the CIF's operation.

### Exclusion criteria

(1) The participants who resigned from their original affiliations during the CIF's operation must be excluded.

(2) The participants who were not in the area during the study period must be excluded.

This research was approved by the Human Research Ethics Committee of Walailak University according to protocol number WUEC-22-130-01, issued on 29 April 2022. The study was conducted following the Declaration of Helsinki. Written informed consent was obtained from all study subjects before participating in the study.

## Data collection

Two questionnaires were used to collect data from the two participant groups, the founding and operating staff, between 27 May – 5 October 2022. These questionnaires were developed based on the guidelines for establishing community isolation during the outbreak of COVID-19 [24], with added preparedness checklists and implementation criteria for a more comprehensive assessment. The evaluation of the questionnaires using the Item-objective Congruence Index (IOC) revealed a content validity ranging from 0.83–0.95, indicating a high level of validity (more than 0.5). The questionnaires' contents are briefly described below.

## CIF preparedness

The questionnaire for the founding staff consisted of three parts, i.e., a description of the study subjects, information about the CIFs, and administrative preparedness of CIFs. Firstly, the description of the subjects comprised of gender, age, educational background, affiliation, job position, and related experience in the field of environmental health. Secondly, information about the CIFs included name, date of establishment, regulatory agency, place of operation, number of beds, food management practice, and air ventilation pattern. Lastly, a preparedness checklist contained four aspects, including 1) location and infrastructure (9 items), 2) staff (2 items), 3) material supplies (5 items), and 4) environmental health preparedness (8 items). The checklist was designed based on the guidelines for establishing CIFs the Department of Health developed. Out of the seven elements in the guideline for CIF preparedness, four were included: 1) administration, 2) location setting and infrastructure, 3) food supply, and 4) infectious waste management and sanitation. The three excluded elements included general facilities and medical equipment, health promotion activities for patients, and surveillance systems. However, specific issues related to surveillance systems, such as wastewater and excreta management, were included in the checklist. Some elements were excluded to align the checklist with the study's objectives and make it more concise for data collection. Additionally, some issues in the checklist, which could have led to uncertainty in judgment by the participants, were modified by adding reference values, such as the adequacy of toilets. Moreover, other issues were added to ensure the checklist had more comprehensive content regarding environmental health management. There were only two answers for each question, i.e. yes or no, for which the authors determined the scoring criteria as 1 or 0, respectively. The scores were interpreted as preparedness levels.

## Environmental health implementation

The questionnaire for the operating staff consisted of two parts, i.e., a description of the study subjects and environmental health implementation. Firstly, the description of the subjects comprised of gender, age, educational background, affiliation, job position, and related experience in the field of environmental health. Secondly, environmental health aspects consisted of six categories, including food sanitation (5 items), infectious waste management (8 items), wastewater and excreta management (5 items), air ventilation (1 item), indoor hygiene (4 items), and personal hygiene (7 items). The scores were interpreted as practice levels by using a 3-point Likert scale [25]. There were three choices for each question according to levels of behavior, i.e. never, sometimes, and always, for which the authors determined the scoring criteria as 0, 1, or 2, respectively. The scores were interpreted as practice levels.

## Data analysis

Descriptive statistics were used to analyze the arithmetic mean and standard deviation of numeric data, as well as the frequency of observations represented as percentages for nominal data. The scores were interpreted to assess preparedness and implementation levels using a criteria-based rating scale based on Bloom's cut-off categories: poor (less than 60%), moderate (60–79%), and good (80–100%) [26,27].

## Results

### Description of the CIFs

Table 1 presents information about the CIFs included in this study. The data showed that most of the CIFs (80.7%) in Nakhon Si Thammarat Province were established by LGOs. Most of the CIFs in this study were situated in school buildings (57.9%), while the rest (42.1%) were in government buildings such as community centers and learning centers. Approximately one–half of the CIFs (45.6%) were classified as small CIFs (1–38 beds). The majority (93.0%) of the CIFs in this study did not use air conditioning and relied on natural ventilation.

### Demographic profile of the participants

Information about the founding and operating staff is presented in Table 2. Most of the participants in both groups were female and held a bachelor's degree. Interestingly, almost half of the founding staff (47.4%) had a master's degree, which was higher than the proportion of the operating staff (21.1%). Additionally, two operating staff members did not possess any degree. While 50.0% of the founding staff had a bachelor's degree in health science, such as public health, environmental health, nursing, and animal science, the operating staff had a much higher proportion (90.7%). In contrast, the majority of the master's degrees in both groups were in humanities and social science programs, such as political science, law, public administration, and business administration. Furthermore, most of the founding staff (89.5%) were officers from LGOs, whereas the operating group was more evenly split between LGOs (49.1%) and public health offices (50.9%). Importantly, most staff members in both groups did not have experience in environmental health.

### Preparedness for establishment of CIFs

**Location and infrastructure.** The majority of the CIFs were located away from the community, with 94.7% situated in such locations as shown in Table 3. In addition, Most of the CIFs (86.0%) provided separated bathrooms and restrooms for male and female patients, with ventilation fans being unavailable in one–third of the CIFs (33.3%). However, air outlets for circulation were presented in most of the CIFs (94.7%). Moreover, most of the CIFs (93.0%) had patient rooms separated from other areas and well-ventilated at 98.2%. Regarding waste management, all the CIFs had storage tanks for solid waste collection. Furthermore, most of the CIFs (71.9%) employed an on-site wastewater treatment system,

**Table 1. Description of the CIFs (n = 57).**

| Description | n (%) |
|---|---|
| Host | |
| Local Government Organizations | 46 (80.7) |
| District Public Health Office | 7 (12.3) |
| District Office | 4 (7.0) |
| Type of the facility | |
| School building/ Children's center | 33 (57.9) |
| Government building | 24 (42.1) |
| Size of the facility (Number of patient bed) | |
| Small (1–38 beds) | 26 (45.6) |
| Medium (39–77 beds) | 16 (28.1) |
| Large (More than 77 beds) | 15 (26.3) |
| Air ventilation system | |
| Natural ventilation (Open air structure) | 53 (93.0) |
| Air conditioning/ natural ventilation | 4 (7.0) |

**Table 2. Demographic characteristics of the participants.**

| Demographic data | Founding staff (n = 57) | Operating staff (n = 57) |
|---|---|---|
| | n (%) | n (%) |
| Gender | | |
| Male | 20 (35.1) | 16 (28.1) |
| Female | 37 (64.9) | 41 (71.9) |
| Highest education level | | |
| High school | 0 (0.0) | 2 (3.5) |
| Bachelor's degree | 30 (52.6) | 43 (75.4) |
| Master's degree | 27 (47.4) | 12 (21.1) |
| Graduated level and program | | |
| High school | | |
| Science and mathematic | 0 (0.0) | 2 (100.0) |
| Bachelor's degree | | |
| Health science | 15 (50.0) | 39 (90.7) |
| Humanities and social Science | 11 (36.7) | 4 (9.3) |
| Natural sciences and technology | 4 (13.3) | 0 (0.0) |
| Master's degree | | |
| Health science | 4 (14.8) | 3 (25.0) |
| Humanities and social science | 23 (85.2) | 9 (75.0) |
| Affiliation | | |
| Local government organization | 51 (89.5) | 28 (49.1) |
| Public health office | 6 (10.5) | 29 (50.9) |
| Related experience in environmental health | | |
| Yes | 20 (35.1) | 22 (38.6) |
| No | 37 (64.9) | 35 (61.4) |

including a septic tank for toilets, a grease trap for kitchen and domestic wastewater, and a collective system. However, only one–third of the CIFs (33.3%) had a system for disinfecting wastewater before discharging.

## Staff

The results (Table 3) indicated that about three-fourths of the CIFs (75.4%) had a sufficient number of staff and volunteers for operation. Most of the founding staff (68.4%) reported that they had access to qualified staff with experience in environmental health. However, when the operating staff themselves were surveyed, only 38.6% reported having the environmental health experience.

## Material supplies

The majority of the CIFs (96.5%) provided full personal protective equipment (PPE), including N95 respirator, face shield, gloves, and coveralls, for all staff. However, a small proportion of the CIFs (3.5%) were unable to afford certain PPE items, such as coveralls. All CIFs received support for other supplies, including appropriate waste storage bins, disinfectants, and cleaning products.

## Environmental health preparedness

The majority of CIFs had undertaken preparations for implementing the environmental health in most of the criteria. However, there were 17.5% of the CIFs reported that their food preparation staff or outsourced food services were not

**Table 3. Preparedness during establishment of CIFs (n = 57).**

| Dimensions | Yes | No |
|---|---|---|
| | n (%) | n (%) |
| 1. Location and infrastructure | | |
| *Location* | | |
| • Away from community, market, or houses[a] | 54 (94.7) | 3 (5.3) |
| *Bathrooms and restrooms* | | |
| • Adequacy (10 patients/ bathroom or restroom)[b] | 49 (86.0) | 8 (14.0) |
| • Ventilation fan[a] | 38 (66.7) | 19 (33.3) |
| • Air inlet and outlet in all rooms[a] | 54 (94.7) | 3 (5.3) |
| *Patient ward/ rooms* | | |
| • Clearly separated from administrative areas and others[a] | 53 (93.0) | 4 (7.0) |
| • Open air and/ or air conditioning with mechanical ventilation[b] | 56 (98.2) | 1 (1.8) |
| *Waste management* | | |
| • Storage tank for solid waste[a] | 57 (100.0) | 0 (0.0) |
| • Wastewater treatment system that function normally[a] | 41 (71.9) | 16 (28.1) |
| • System for disinfecting wastewater before discharging[a] | 19 (33.3) | 38 (66.7) |
| 2. Staff | | |
| • Sufficient staffs and volunteers for operation[b] | 43 (75.4) | 14 (24.6) |
| • Qualified operating staff who understand environmental health or environmental management[b] | 39 (68.4) | 18 (31.6) |
| 3. Material supplies for staffs, volunteers, and/or patients | | |
| • Personal protective equipment (PPE)[a] | 55 (96.5) | 2 (3.5) |
| • Waste storage bins that were completely closed and convenient to move[a] | 57 (100.0) | 0 (0.0) |
| • Waste storage bins with red plastic bags for infectious waste[a] | 57 (100.0) | 0 (0.0) |
| • Alcohol or disinfectant[b] | 57 (100.0) | 0 (0.0) |
| • Cleaning products for utensils and surfaces[b] | 57 (100.0) | 0 (0.0) |
| 4. Environmental health preparedness | | |
| • Food and drinking water supplied 3 meals a day[a] | 56 (98.2) | 1 (1.8) |
| • Drinking water that is sealed bottled water[b] | 57 (100.0) | 0 (0) |
| • Food preparation staffs or outsources that follow food sanitation principles[b] | 47 (82.5) | 10 (17.5) |
| • Authorized or government agency that collected, transported, and transferred of infectious waste[a] | 57 (100.0) | 0 (0.0) |
| • Procedures for managing solid waste[b] | 56 (98.2) | 1 (1.8) |
| • Protocol for providing knowledge in environmental health management to staffs and patients[b] | 43 (75.4) | 14 (24.6) |
| • Measures for cleaning various areas of the CIF[b] | 55 (96.5) | 2 (3.5) |
| • Measures to prevent pathogen contamination to the community[a] and/or allow visiting only authorized person[b] | 50 (87.7) | 7 (12.3) |

[a]Preparedness guideline presenting in the guidelines developed by Department of Health (2021).

[b]Additional preparedness issues proposed by the authors

following proper food sanitation practices. Additionally, one-fourths of the CIF (24.6%) had not established procedures for educating their staff and patients about environmental health.

## Levels of the preparedness for establishment of CIFs

The majority of the CIFs in this study were well-prepared in terms of staff (61.4%), supplies (100%), and environmental health readiness (82.4%). However, most of the CIFs (61.4%) had a moderate level of preparedness in terms of location and infrastructure. Overall, the preparedness level for all four dimensions in most of the CIFs in this study (75.4%) was good (Table 4).

**Table 4. Levels of preparedness for establishment of CIFs (n = 57).**

| Dimensions | Level of preparedness: n (%) | | |
|---|---|---|---|
| | Poor | Moderate | Good |
| 1. Location and infrastructure | 4 (7.0) | 35 (61.4) | 18 (31.6) |
| $\bar{x}$= 7.1, S.D. = 1.26, max = 9.0, min = 4.0) | | | |
| 2. Staff | 22 (38.6) | 0 (0.0) | 35 (61.4) |
| $\bar{x}$= 1.4, S.D. = 0.78, max = 2.0, min = 0.0) | | | |
| 3. Material supplies for staffs, volunteers, and/or patients | 0 (0.0) | 0 (0.0) | 57 (100.0) |
| $\bar{x}$= 5.0, S.D. = 0.18, max = 5.0, min = 4.0) | | | |
| 4. Environmental health preparedness | 1 (1.8) | 9 (15.8) | 47 (82.4) |
| $\bar{x}$= 7.4, S.D. = 0.99, max = 8.0, min = 4.0) | | | |
| All dimensions | 1 (1.8) | 13 (22.8) | 43 (75.4) |
| $\bar{x}$= 20.8, S.D. = 2.46, max = 24.0, min = 13.0) | | | |

## Environmental health implementation in CIFs

Table 5 presents information on the implementation of environmental health issues in CIFs, including food sanitation, infectious waste management, wastewater and excreta management, air ventilation, indoor hygiene, and personal hygiene.

## Food sanitation

All the CIFs provided food for the patients from external sources. However, 11 out of the 57 CIFs (19.3%) prepared the food (Table 5). Most of the CIFs' food providers consistently adhered to good practices for food sanitation. These practices included 100.0% compliance with hand washing and wearing face mask, as well as 94.7% compliance with fresh cooking, 89.5% compliance with separating side dishes from rice in boxed food, and 98.2% compliance with proper storage of cooked food, where the food was placed at least 60 cm from the ground. Cooks and food handlers in some CIFs (36.4%) did not always wear clean clothing and hair restraints and refrained from wearing jewelry. Regarding food suppliers, approximately half of the CIFs (49.1%) consistently provided food from certified food shops, while the other half (49.1%) did so occasionally. In this study, the majority of the CIFs (54.4%) occasionally served food containing coconut milk, high-fat content, and raw ingredients.

## Infectious waste management

In Table 5, it was observed that most CIFs followed good practices in managing infectious waste. Nonetheless, there were two issues that need attention. Firstly, some CIFs (26.3%) did not consistently inform patients to collect their waste in tightly closed double bags and spray them with disinfectant. Regular announcements and individual notices to patients who did not follow proper waste collection practices were not fully implemented. Secondly, some CIFs (15.8%) did not maintain complete closure of waste containers in the waste shelter or use sufficient containers to avoid leftover waste.

## Wastewater and excreta management

The evaluation of environmental health implementation revealed significant dissatisfaction with wastewater and excreta management. Some CIFs did not conduct regular check on the wastewater system (36.8%) and the sewage storage system and septic tank (17.5%). Moreover, some CIFs did not periodically check the disinfecting systems (87.7%), and the free residual chlorine in the effluent before discharge (89.5%). Additionally, the majority of the CIFs (77.2%) never determined the quality of the wastewater discharge at least once a month.

**Table 5. Environmental health implementation in CIFs (n = 57).**

| Dimensions | Level of practice: n (%) | | |
|---|---|---|---|
| | **Never** | **Sometimes** | **Always** |
| Food sanitation | | | |
| *Food provided by the CIFs (n = 11): Cooks and food handlers* | | | |
| • Clean clothes and apron, boots/ shoes, and no jewelry | 0 (0) | 4 (36.4) | 7 (63.6) |
| • Proper hair restraints such as a hair net | 0 (0) | 4 (36.4) | 7 (63.6) |
| • Hand washing thoroughly with soap, short fingernails, and no nail polish | 0 (0) | 0 (0) | 11 (100) |
| • Wearing of face mask | 0 (0) | 0 (0) | 11 (100) |
| *Food provided by outsources (n = 57)* | | | |
| • Food providers were certified by a local government agency or public health organization | 1 (1.8) | 28 (49.1) | 28 (49.1) |
| • Freshly cooked food (not more than 3–4 hours) before serving | 0 (0) | 3 (5.3) | 54 (94.7) |
| • Food cooked with no coconut milk, high fat content, and raw ingredients | 2 (3.5) | 31 (54.4) | 24 (42.1) |
| • Rice was separated from side dish | 0 (0) | 6 (10.5) | 51 (89.5) |
| • No packed food leaved on the floor, keep food away from sunlight and high temperature area | 0 (0) | 1 (1.8) | 56 (98.2) |
| Infectious waste management | | | |
| • Announcement to all patients how to manage their own waste | 0 (0) | 1 (1.8) | 56 (98.2) |
| • Distribution of well closed infectious waste storage bin with red plastic bag to all patients | 0 (0) | 2 (3.5) | 55 (96.5) |
| • Announcement to all patients for collecting their own waste in tightly closed double bags, top sprayed with disinfectant | 0 (0) | 15 (26.3) | 42 (73.7) |
| • Workers responsible for collecting, transporting, and transferring of infectious waste have undergone training | 1 (1.8) | 5 (8.8) | 51 (89.5) |
| • CIF staffs wear appropriate PPE[a] while handling infectious waste | 1 (1.8) | 2 (3.5) | 54 (94.7) |
| • Infectious waste shelter with completely closed waste containers and no leftover waste | 0 (0) | 9 (15.8) | 48 (84.2) |
| • Infectious waste shelter with one-way entrance and notification sign | 1 (1.8) | 2 (3.5) | 54 (94.7) |
| • Infectious waste was treated with standardized procedure, conducting by authorized government or private agencies | 0 (0) | 5 (8.8) | 52 (91.2) |
| Wastewater and excreta management | | | |
| • Periodic checking of the wastewater system[b] | 21 (36.8) | 12 (21.1) | 24 (42.1) |
| • Periodic checking of disinfecting system[b] | 50 (87.7) | 2 (3.5) | 5 (8.8) |
| • Determination of free residual chorine (< 1 mg/L) in the effluent before discharging 2 times/day[b] | 51 (89.5) | 3 (5.3) | 3 (5.3) |
| • Periodic checking of sewage storage system and septic tank | 10 (17.5) | 19 (33.3) | 28 (49.1) |
| • Determination of wastewater discharge quality at least once a month | 44 (77.2) | 7 (12.3) | 6 (10.5) |
| Air ventilation | | | |
| *Open air (n = 57)* | | | |
| • Air inlet and outlet at least two channels from outside to let the air flow through the building (cross ventilation) | 1 (1.8) | 1 (1.8) | 55 (96.5) |
| *Open air and air conditioner (n = 4)* | | | |
| • Open ventilation fan to suck air out of the building | 0 (0) | 1 (25.0) | 3 (75.0) |
| • Opened doors and windows for ventilation at least 2 hours before turning on the air conditioners | 1 (25.0) | 1 (25.0) | 2 (50.0) |
| Indoor hygiene | | | |
| • Daily cleaning of the CIF with cleaning products and disinfectants | 0 (0) | 6 (10.5) | 51 (89.5) |
| • Daily cleaning of the bathroom and restrooms with cleaning products and disinfectants | 0 (0) | 8 (14.0) | 49 (86.0) |
| • Daily cleaning of the surfaces that are frequently contacted with cleaning products and disinfectants (e.g. doorknobs and windows) | 0 (0) | 5 (8.8) | 52 (91.2) |
| • Open the doors and windows for sunlight in the patient room | 0 (0) | 11 (19.3) | 46 (80.7) |
| Personal hygiene | | | |
| • Announcement to all patients to wear a face mask all the time | 0 (0) | 5 (8.8) | 52 (91.2) |
| • Announcement to all patients for rules and regulations | 0 (0) | 3 (5.3) | 54 (94.7) |

*(Continued)*

**Table 5.** (Continued)

| Dimensions | Level of practice: n (%) | | |
|---|---|---|---|
| | **Never** | **Sometimes** | **Always** |
| • Preparation of soap and alcohol gel to be ready for use | 0 (0) | 2 (3.5) | 55 (96.5) |
| • Wearing of PPE during work in the patient area | 0 (0) | 2 (3.5) | 55 (96.5) |
| • Wash hands frequently or every time after handling things | 0 (0) | 2 (3.5) | 55 (96.5) |
| • Separated restrooms for staffs and patients | 0 (0) | 2 (3.5) | 55 (96.5) |
| • Cleanse the body and change work clothes before leaving the CIF | 1 (1.8) | 22 (38.6) | 34 (59.6) |

[a]PPE: rubber gloves; boots; apron; N95 face mask; goggles; face shield; coverall

[b]The practice level of CIFs where wastewater treatment system or disinfecting system was not available was "Never".

## Air ventilation

The majority of the CIFs (96.5%) used natural ventilation and always opened both the air inlet and outlet to allow cross ventilation (i.e., open doors and windows), which is better than single-sided ventilation (i.e., open windows only). However, out of the four CIFs that used air conditioning, one CIF (25%) never opened doors and windows at least two hours before turning on the air conditioners. In addition, only one CIF (25%) occasionally used a ventilation fan to extract air from the building.

## Indoor and personal hygiene

The majority of the CIFs conducted daily cleaning (89.5%) using cleaning products and disinfectants in the bathroom and restrooms (86.0%), as well as on frequently touched surfaces (91.2%), such as toilet seats, toilet covers, toilet flushers, doorknobs or latches and faucets, and windows. Furthermore, most of the CIFs (80.7%) usually opened the doors and windows to let sunlight into the patient's room. For personal hygiene, most of the CIFs always maintained good practices. Nonetheless, some operating staff (38.6%) did not always shower and change work clothes before leaving the CIFs.

## Levels of environmental health implementation

The overall assessment for environmental health implementation in CIFs was interpreted as practice levels, i.e., poor, moderate, and good, according to the score for each issue as shown in Table 6. The data indicated that the CIFs performed well in most of the environmental health dimensions, including food sanitation (84.2%), infectious waste management (96.5%), air ventilation (96.5%), indoor hygiene (87.7%), and personal hygiene (96.5%). However, the management of wastewater and excreta was a major issue of concern, with most of the CIFs (93.9%) scoring poorly in this category. As a result, the overall assessment of the CIFs based on all the criteria showed moderate and good levels for 50.9% and 47.3% of the CIFs, respectively.

## Discussion

Thailand has a unique approach to managing COVID-19 using a decentralized governance model. The decentralized governance model allows LGOs to make decisions, manage resources and services, and respond to their area's needs. Therefore, this model could facilitate rapidly managing various crises, including the COVID-19 pandemic. The approach of aligning responsibility to LGOs for establishing CIFs (see Fig 1) solved the problem of bed shortages for treating hospital patients. Compared to other countries, they employed centralized strategies, and national health authorities played a significant role in the decision-making and allocating of resources, which differed from Thailand. This study showed that Thailand's decentralized governance model facilitated small CIFs' operation in local communities, with LGOs playing a significant role (see Table 1).

Table 6. Levels of environmental health implementation in CIFs (n = 57).

| Dimensions | Levels of management: n (%) | | |
|---|---|---|---|
| | Poor | Moderate | Good |
| 1. Food sanitation | 0 (0.0) | 9 (15.8) | 48 (84.2) |
| ($\bar{x}$ = 8.7, S.D. = 1.05, max = 10, min = 6) | | | |
| 2. Infectious waste management | 0 (0.0) | 2 (3.5) | 55 (96.5) |
| ($\bar{x}$ = 15.2, S.D. = 1.20, max = 16, min = 10) | | | |
| 3. Wastewater and excreta management | 53 (93.0) | 4 (7.0) | 0 (0.0) |
| ($\bar{x}$ = 2.7, S.D. = 1.81, max = 6, min = 0) | | | |
| 4. Air ventilation | 2 (3.5) | 0 (0.0) | 55 (96.5) |
| ($\bar{x}$ = 2.0, S.D. = 0.29, max = 2, min = 0) | | | |
| 5. Indoor hygiene | 2 (3.5) | 5 (8.8) | 50 (87.7) |
| ($\bar{x}$ = 5.6, S.D. = 0.80, max = 6, min = 3) | | | |
| 6. Personal hygiene | 2 (3.5) | 0 (0.0) | 55 (96.5) |
| ($\bar{x}$ = 13.3, S.D. = 1.27, max = 14, min = 7) | | | |
| All dimensions | 1 (1.8) | 29 (50.9) | 27 (47.3) |
| ($\bar{x}$ = 47.4, S.D. = 3.99, max = 54, min = 31) | | | |

Strong underlying health systems, especially in primary healthcare, are the drivers to identify and contain COVID-19 infections and outbreaks. Village health volunteers, ordinary people who devoted themselves to helping those in their village, were recognized as an essential component of the national COVID-19 response. In CIFs, these volunteers were key in screening and inspecting infected individuals in their communities, informing community leaders, and referring patients for timely treatment or appropriate containment by HI or CI when necessary. Promptly responding to asymptomatic and mildly symptomatic COVID-19-confirmed patients is crucial for effective surveillance, early detection, and health monitoring of COVID-19 patients in Thailand.

CIFs were established within pre-existing public infrastructures to rapidly response to the outbreak situation. Repurposed facilities for CI that have been reported included, for example, gymnasiums, conventions, exhibition centers [28], training centers, resorts, and dormitories [11]. Schools and government buildings were the pre-existing public infrastructures repurposed to establish CIFs in this study. Most were open-air structures capable of diluting infectious aerosols through ventilation with outside air, thus reducing pathogens accumulation in cases where mechanical ventilation was unavailable. Open-air buildings are usually present in rural communities because of low investment and electricity costs. The installation of additional equipment would delay the immediate operation of the CIFs during an outbreak.

In terms of capacity (number of beds), all CIFs in this study were relatively small compared to facilities utilizing the CI strategy in other countries, such as a human resource development center (100 beds) in South Korea [12], bungalows in a holiday camp (250 single rooms) in Hong Kong [29], and a convention hall (3200 beds) in Singapore [30]. The sizes or numbers of beds in each CIF might vary depending on factors such as the number of patients in each community willing to be admitted, the capacity of the infrastructure, and available resources, including financial support from LGOs, donations, and supporting staff. Most CIFs in Thailand, except those in the Bangkok Metropolitan Region, operated under decentralized governance through LGOs. As a result, small CIFs predominated in this study.

The overall preparedness for establishment of CIFs was good, while the preparedness in terms of location and infrastructure was moderate in most of the CIFs. Data from Table 3 indicates that the absence of a wastewater treatment system—particularly one for disinfection prior to discharge—substantially lowered the overall preparedness scores for Location and Infrastructure to a moderate level. This situation may be because the local CIFs in this study were established within existing community infrastructure, as previously mentioned, to ensure a swift response in accommodating patients

during an urgent situation. The priority was to isolate infected individuals from their families and the community quickly. However, these local facilities often lacked adequate spatial readiness as they were not initially designed for patient habitation. Therefore, to prevent the spread of disease in similar future outbreaks, relevant agencies such as LGOs should prepare suitable sites or improve existing infrastructure. This process should involve a comprehensive assessment of their suitability, covering environmental health aspects, to ensure a timely and effective response to potential future pandemics.

Most CIFs were located at least ten meters far away from adjacent buildings or communities, as recommended by the Department of Health [24] to prevent the spread of the virus. The WHO reported the occurrence of aerosol transmission of SARS-CoV-2 [31]. A study conducted in a hospital in Wuhan (China) showed that SARS-CoV-2 was detected in 15% of the collected outdoor samples at 10 meters from the doors of in-patient and out-patient buildings while it was not detected in residential and open public areas [32]. Therefore, transmission to nearby communities is probably less concerning than nosocomial infections inside CIFs. A well-planned location for CIFs would make it very unlikely for the COVID-19 virus to spread to the community.

CIFs should be mainly separated into three main areas, i.e. 1) patients' area, 2) clean area for staff, and 3) area for system administration and utilities [24]. Patient wards in most of the CIFs (98.2%) operated in open-air or well-ventilated spaces. For CIFs that did not operate in open space and air conditioners were used, mechanical ventilation was installed to prevent air recirculation and reduce microbial accumulation. Sufficient bathrooms and restrooms not only made the patients comfortable but also prevented waste accumulation and maintain cleanliness. Insufficient ventilation has been shown to pose a high risk of cross-infection in public toilets [33]. In case of inadequacy in this study (14.0% of the CIFs), newly built or mobile restrooms should be provided. Additionally, ventilation fan in the bathroom or restroom was not available in one–third of the CIFs; however, the air outlets existed in all rooms of most CIFs. For the preparedness in terms of waste management, all CIFs had storage tanks for solid waste collection and authorized agencies for infectious waste collection and transfer for appropriate treatment. It is worth noting that wastewater treatment and disinfecting system were not available in 28.1% and 66.7% of the CIFs, respectively. For CIFs without a treatment system, a disinfection system before releasing wastewater into drainage channels or public areas was required [24].

Interestingly, the level of preparedness in terms of staff was poor in some CIFs (38.6%). This finding may be because, according to Table 3, the assessment of staff preparedness by founding staff was based on two criteria: 1) the sufficiency of staff and volunteers for CIF operations, and 2) the presence of qualified operating staff with an understanding of environmental health management. Regarding the first criterion on personnel numbers, during the CIFs' establishment phase, some LGOs, particularly smaller ones, had limited staff and had not yet established collaborations with other agencies to supplement their workforce. As for the operating staff, some founding staff (31.6%) reported that they were not qualified. However, self-reported educational backgrounds by the operating staff show that a majority (90.7%) held degrees in health sciences. A closer look at individual data revealed that only one person had a degree in environmental health. Furthermore, Table 2 indicates that the majority (61.4%) had no prior experience in environmental health. This data suggests a shortage of personnel specializing in environmental health within LGOs. It is essential to recruit more of these specialists for environmental health operations, especially if LGOs are to continue their role in establishing CIFs for future pandemic response. These measures would ensure the availability of qualified personnel capable of managing the environment to control infectious disease outbreaks effectively.

The founding staff should possess adequate knowledge and experience to make decisions that consider all essential aspects from an environmental health perspective. The different patterns in the educational background between the founding and operating staff might be related to the organization to which they belong. Most founding staff (89.5%) were officers from LGOs, while more than half of the operating staff (50.9%) came from public health agencies. The findings of this study reflect the shortage of qualified personnel suitable to be founding staff of CIFs, which may result from some LGOs lacking departments directly involved in environmental health work. The cooperation from the Public Health Office in supporting staff participation as operating staff in CIFs was part of the effort to maintain appropriate practices in CIFs.

However, operations requiring assistance and consultation from the Public Health Office might cause delays in responding to urgent situations.

Moreover, the result also showed that most of the Master's degrees in both groups of participants were in humanities and social science fields, which might be because the postgraduate degree was beneficial for their promotion to executive positions. The founding staff, mostly from LGOs, showed a higher proportion of Master's degree holders (47.4%) than the operating staff (21.1%). The founding staff was responsible for checking the readiness of the infrastructure and personnel and setting up the systems. The postgraduate degree would benefit the management of resources and the decision-making of the founding staff. However, as already mentioned, the result showed that most of the founding staff lacked relevant knowledge and experience in environmental health.

The operation of CIFs relied on more than just infrastructure and personnel; material supplies were also essential. All the CIFs in this study were well-prepared regarding supplies at a good level (see Table 4). LGOs maintain a reserve or emergency budget that can be utilized if an area is declared a disaster zone under the authority of the local government council by the decentralized structure of the Thai administrative system. This approach enables each CIF to secure sufficient funding to allocate various supplies required for operations. Furthermore, donations of essential goods, food, and medical supplies are provided to each local CIF by the local population and private organizations.

Although the overall preparedness in terms of supplies is at a good level, there is still a small number of CIFs (3.5%) that did not have full PPE for working with COVID-19 patients. The shortage was attributed to a high demand for N95 respirators and coveralls during the pandemic. As an alternative, medical masks were used where necessary, and raincoats were used as isolation gowns during the coverall shortage. Other material supplies required for maintaining proper waste management and environmental sanitation, such as waste storage bins, disinfectants, and cleaning products for utensils and surfaces, were available in all CIFs. These supplies were not only supported by the host organization but also by donations from private sectors and individuals.

The CIFs had prepared for environmental health management in most of the criteria. There were some CIFs (17.5%) where food preparation staff or outsource suppliers did not follow proper food sanitation practices. Food supplied by outsources or unknown sources by donation could not be monitored to ensure the sanitation practice. It is important to establish procedures for setting up a list of qualified or certified food suppliers and protocols for inspecting food from unknown sources. In addition, some CIFs (24.6%) did not establish procedures for educating staff and patients about environmental health management. In these cases, the environmental health management issue was not prioritized for patients, there were limitation in terms of staff knowledge and experience.

The data from the operating staff on environmental health implementation indicated poor management of wastewater and excreta. This finding is consistent with the lack of wastewater treatment and disinfection systems in the infrastructure of CIFs. The effluent should be disinfected by adding chlorine before discharging into the environment. Besides the availability of infrastructures, limitations in the knowledge and experience of the staff to operate the system were suggested (see Table 2). It is important to note that SARS-CoV-2, compared to non-enveloped human enteric viruses, is less stable outside the host when released into the environment, potentially limiting public health concerns about its transmission. Nevertheless, transmission of COVID-19 and SARS through sewage has been reported in studies in Hong Kong [34,35]. In addition, available evidence suggests that wastewater treatment or proper maintenance of on-site sanitation systems is crucial to reduce the risk posed by SARS-CoV-2 [36]. Therefore, the discharge of wastewater without appropriate treatment might increase the probability of the disease transmission. Training on wastewater disinfection for the operating staff is needed to prevent the spread of the virus to the environment.

In addition to medical care, nutrition and hygiene are important to help the patient's recovery. Establishing a list of certified food suppliers and protocols for inspecting food from unknown sources is crucial. In some cases, nearby restaurants with short transportation times were selected. Patients should avoid food cooked with coconut milk, high-fat content, and raw ingredients. The data in Table 5 showed that most of the CIFs in this study sometimes served that kind of food to the

patients, for example, coconut milk curry and salad. Coconut milk is easily spoiled within a few hours, especially in warm temperatures. As a result, it is recommended to consume food with coconut milk within a few hours. In addition, high-fat content and raw ingredients also pose health risks for patients with underlying diseases, such as cardiovascular diseases and microbial food poisoning, respectively. This study found challenges in ensuring food safety for food from outside suppliers or donations, as delivery times and food types could not be strictly controlled. Despite these risks, no related foodborne illness cases were reported.

Proper infectious waste collection is important for preventing transmission of the virus to the staff, especially those who were responsible for the waste collection, and for the spread of the virus to the communities and the environment. After packing of the first layer of double red bags, they should be disinfected by top spraying of disinfectants before disposal. It's difficult to observe this behavior, so this procedure should be usually informed, especially for new patients, to ensure proper waste collection. The rate of waste generation is important for estimating the number and size of the waste containers, storage areas, transportation, and waste treatment capacities. Governmental authorities should provide guidelines for CIFs to estimate the required capacity from the establishment stage. Besides, appropriate adjustment of the capacity to serve with fluctuation of the patient number during operation by well-trained staff is necessary.

Air ventilation is an important determinant for setting up CIFs to prevent airborne transmission. The ventilation rate can be improved by different means, such as opening windows or using air inlets/outlets, as well as installing or adjusting ventilation fans. If the ventilation fan does not work, natural ventilation or administrative controls such as temporarily limiting the number or residence time of the occupants may be alternatives to maintain adequate ventilation rate [37]. The type of ventilation applied depended on the availability of the pre-existing infrastructure and the operational cost. Installing additional equipment might delay the immediate operation of the CIFs in the outbreak. However, certain minimum requirements should be achieved, such as ensuring air flow direction from clean to less clean area, and that air is exhausted directly to the outside and away from air intake. Data in Table 5 showed that one-fourth of the CIFs operated using air conditioners only applied the ventilation occasionally. This practice should be regularly performed to avoid the accumulation of the virus inside the building.

To maintain a safe indoor environment for patients and staff, it is important to ensure regular cleaning of the patients' rooms and common areas. Studies have shown that the SARS-CoV-2 virus can remain viable on surfaces, such as plastic, stainless steel, wood, cloth, and glass, for up to 9 days [38–40]. Finger contact with virus-contaminated surfaces and subsequent finger contact with the facial membranes could lead to the SARS-CoV-2 infection. Open the doors and windows for sunlight in the patient room were conducted in most of the CIFs. It has been documented that SARS-CoV-2 on surfaces is inactivated rapidly with sunlight [41,42]. Besides, it has been shown that viruses transmitted from contaminated clothes to uncontaminated ones [43]. The operating staff enrolled in this study did not enter the patients' area which was separated from the staff area. Food delivery, waste collection, cleaning, and maintenance were carried out by other staff under their supervision. In addition, online communication via mobile applications such as LINE was used instead of face-to-face communication. Moreover, the staff was accommodated separately from their families during the period they were on duty. The staff might not be concerned about this practice because they did not have direct contact with the patients.

This research could not be conducted during the outbreak period, when the staff were working, due to restrictions on access to CIFs. Additionally, the staff had to be quarantined because of their close contact with infected individuals and could not meet others. As a result, the researchers could not collect data on the environment within the CIFs during the operation and had to rely on retrospective data from the staff after the outbreak. Therefore, providing suggestions or feedback for improvements during the operational period was impossible. Furthermore, there was a lack of specific data on the number of infected individuals in each community, making it difficult to assess the effectiveness of CIFs' operations.

However, the overall infection data for the province reveals an interesting trend. Fig 2 illustrates the number of confirmed cases in Nakhon Si Thammarat Province during the 2021–2022 pandemic. The data showed that CIFs were established

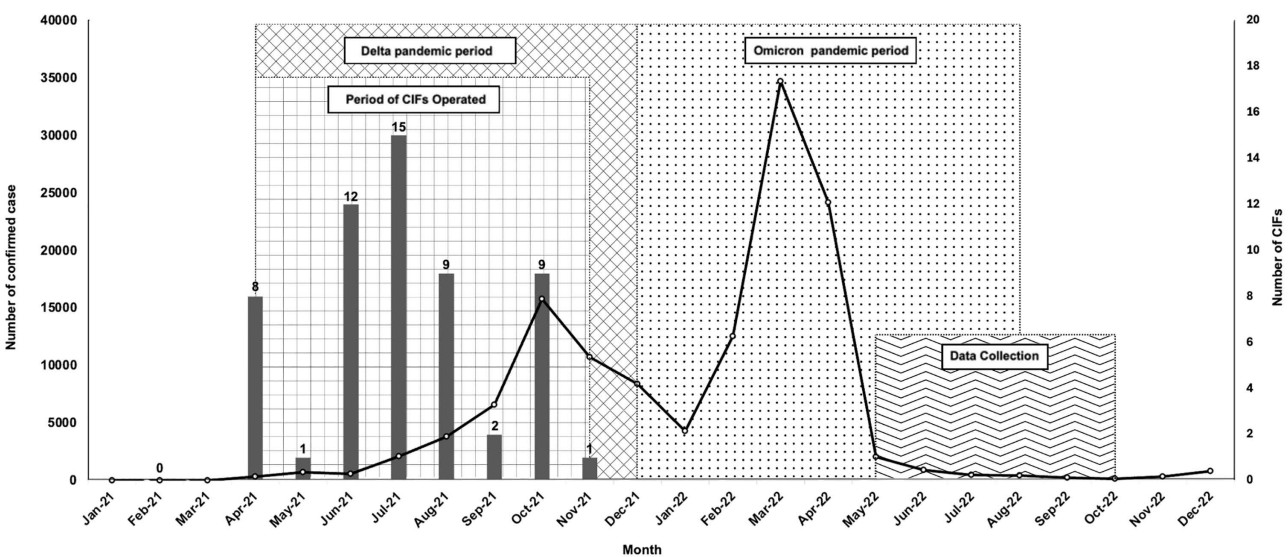

**Fig 2. A relationship between number of CIFs and confirmed COVID-19 cases in Nakhon Si Thammarat Province during the pandemic in 2021–2022.** ▇ represents the number of CIFs newly established in the month. ─○─ represents the number of new COVID-19 confirmed cases in the month. ▯ represents the CIF operation period. ▨ represents the Delta variant-dominant period. ▦ represents the Omicron variant-dominant period. ▧ represents the data collection period.

during the Delta period (April – November 2021). In April 2021, cases of the Delta strain started to appear in many areas, and the number gradually increase until reaching the peak in October. It should be noted that the number of confirmed cases was consistent with national outbreak patterns. The number of CIFs established increased as the number of infected people increased. It is worth noting that the number of CIFs dramatically increased when there was a large cluster. The number of infected people was the main factor in the decision to close CIFs in each area when the number of infected people decreased. CIFs were discontinued during the downturn of the Delta pandemic period as the government changed the policy to allow patients to quarantine at home or receive treatment in hospitals if at risk or with severe symptoms.

Since the beginning of the Omicron outbreak, the number of confirmed cases had been increased exponentially. Despite widespread vaccination was promoted to boost immunity, the number of confirmed cases infected with the Omicron variant were apparently higher than those during the Delta period. The characteristics of different virus strains might influence the pattern of the outbreak. The Omicron strain is a strain that arose after the outbreak of the Delta strain, which occurred from the mutation of the virus. Although the Omicron variant is less virulent [44], but is approximately 2–3 times more contagious than the Delta variant [45]. Moreover, as shown in Fig 2, the operation of CIFs appeared to delay the peak of Delta cases. The Delta strain took approximately six months from April to October 2021, to reach its peak. In contrast, the Omicron period, without CIFs, took only two months, from January to the peak in March 2022. Compared to the Omicron outbreak, fewer people were vaccinated during the Delta period. It's suggested that without CIFs in response to the Delta outbreak, the number of confirmed cases could have risen even in the early stages.

## Conclusions

The operation of CIFs under the decentralized model, following the guidelines for the establishment of CIFs during the COVID-19 pandemic [24], facilitated effective disease control. The guidelines provided step-by-step procedures for readiness checking and operating CIFs, creating an ideal scenario for implementation based on current scientific evidence and the best allocation of resources. However, this study revealed that in terms of environmental health management, there

were some issues that could not be appropriately implemented, mainly due to limitations of infrastructure and environmental health personnel in the CIFs. The role of environmental health practitioners, who are normally recognized as major contributors to emergency preparedness and response, was not highlighted in this pandemic. LGOs should prepare personnel to be ready for environmental health work by, for example allocating the related positions and organizing training programs. The knowledge of environmental health among LGO staff is valuable not only during a crisis but also for local agencies responsible for public welfare, as environmental health is an important health determinant.

The findings of this study are valuable for improving the preparedness of environmental health management of CIFs. The lessons learned from this study can also be applied by other countries in similar contexts to establish CIFs during future infectious disease outbreaks. Furthermore, there is a need to establish international guidelines for environmental health management in CIFs or non-healthcare facilities. These guidelines should be practical and based on in the limited availability of resources and local constraints during emergencies, while still being effective in preventing the spread of diseases. The nature of future diseases is unpredictable and diverse. However, establishing minimum requirements for environmental health practices based on up-to-date scientific evidence and the lessons learned from the COVID-19 pandemic will better prepare us for future infectious disease outbreaks.

## Supporting information

**S1. Raw data.**
(PDF)

**S2. Inclusivity in global research.**
(PDF)

## Acknowledgments

This article is part of a master's thesis, supported by the Master of Public Health Program, School of Public Health, Walailak University.

## Author contributions

**Conceptualization:** Udomratana Vattanasit, Jira Kongpran, Chamnong Thanapop.

**Data curation:** Arif Cheseng.

**Investigation:** Arif Cheseng.

**Methodology:** Udomratana Vattanasit, Arif Cheseng, Jira Kongpran, Chamnong Thanapop.

**Supervision:** Udomratana Vattanasit.

**Visualization:** Arif Cheseng.

**Writing – original draft:** Udomratana Vattanasit, Arif Cheseng.

**Writing – review & editing:** Udomratana Vattanasit, Jira Kongpran, Chamnong Thanapop, Maisarah Nasution Waras.

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
