## [Decision Letter · Decision Letter 0]

28 Feb 2025

PONE-D-25-02239Environmental health management in local community isolation facilities during COVID-19 pandemic: A case study in Nakhon Si Thammarat, ThailandPLOS ONE

Dear Dr. Vattanasit,

Thank you for submitting your manuscript to PLOS ONE. After careful consideration, we feel that it has merit but does not fully meet PLOS ONE’s publication criteria as it currently stands. Therefore, we invite you to submit a revised version of the manuscript that addresses the points raised during the review process.

We look forward to receiving your revised manuscript.

Kind regards,

Babak Pakbin

Academic Editor

PLOS ONE

Journal Requirements:

Additional Editor Comments : No more comments. 

Reviewers' comments:

Reviewer's Responses to Questions

**Comments to the Author**

1. Is the manuscript technically sound, and do the data support the conclusions?

Reviewer #1: Yes

Reviewer #2: Partly

2. Has the statistical analysis been performed appropriately and rigorously? 

Reviewer #1: Yes

Reviewer #2: I Don't Know

3. Have the authors made all data underlying the findings in their manuscript fully available?

Reviewer #1: Yes

Reviewer #2: Yes

4. Is the manuscript presented in an intelligible fashion and written in standard English?

Reviewer #1: Yes

Reviewer #2: No

5. Review Comments to the Author

Reviewer #1: Abstract

o Specify the time period of the study

o Clarify the total number of staff enrolled

o Brief mention of the statistical methods used

o Consider adding a brief background sentence

o The final statement suggests that guidelines should be adopted for future disease outbreaks but doesn't specify how they should be improved. Consider adding a phrase like:

o Provide more information about questionnaire

Introduction

o The research gap is stated but could be emphasized more clearly.

o The objectives should be stated more clearly and concisely.

Methods

o Clarify the sampling method

o Provide more details on how the 57 CIFs were selected from the total 114

o Provide the process of developing questionnaires based on guidelines

o Provide rationale for the specific score ranges and cut-off points

o Clarify if any statistical software was used for analysis

Results

o Explain the practical significance of some percentages

o Include p-values or statistical significance tests where relevant

o Analyze why there are differences in educational backgrounds

Discussion

o Elaborate on Thailand's unique approach to COVID-19 management

o Provide more context about the decentralized governance model

o More exploration of why certain environmental health practices were suboptimal

o Discuss potential systemic reasons for implementation challenges

o Expand on comparisons with CIF strategies in other countries

o Expand on study limitations

o Provide more forward-looking recommendations

Reviewer #2: Dear sir,

It is a valuable study, but some corrections are needed.

Items related to sampling and people's characteristics should be discussed in the method. The discussion is long and some content should be included in the introduction, methodology and results.

6. PLOS authors have the option to publish the peer review history of their article (what does this mean?). If published, this will include your full peer review and any attached files.

Reviewer #1: No

Reviewer #2: **Yes: **prof, Maryam Javadi

---

## [Author Response · Author response to Decision Letter 1]

9 Apr 2025

Thank you for giving me the opportunity to submit a revised draft of my manuscript titled “Environmental health management in local community isolation facilities during COVID-19 pandemic: A case study in Nakhon Si Thammarat, Thailand” to PLOS ONE. I appreciate the time and effort you and the reviewers have dedicated to providing valuable feedback on the manuscript. I am grateful to the reviewers for their insightful comments on my paper.

I have incorporated changes to reflect most of the suggestions provided by the reviewers. The manuscript's changes were shown using track changes in the separated file (File name: PONE-D-25-02239-Revised Manuscript with Track Changes). Point-by-point responses to the reviewer's comments and concerns are presented in a separated file ( File name: PONE-D-25-02239-Response to Reviewers). For your information, the line numbers mentioned in the document refer to the lines in the file with track change, not the unmarked version of the revised manuscript.

---

## [Decision Letter · Decision Letter 1]

31 Aug 2025

PONE-D-25-02239R1Environmental health management in local community isolation facilities during COVID-19 pandemic: A case study in Nakhon Si Thammarat, ThailandPLOS ONE

Dear Dr. Vattanasit,

Thank you for submitting your manuscript to PLOS ONE. After careful consideration, we feel that it has merit but does not fully meet PLOS ONE’s publication criteria as it currently stands. Therefore, we invite you to submit a revised version of the manuscript that addresses the points raised during the review process.

We look forward to receiving your revised manuscript.

Kind regards,

Phuping Sucharitakul

Academic Editor

PLOS ONE

Journal Requirements:

Reviewers' comments:

Reviewer's Responses to Questions

**Comments to the Author**

1. If the authors have adequately addressed your comments raised in a previous round of review and you feel that this manuscript is now acceptable for publication, you may indicate that here to bypass the “Comments to the Author” section, enter your conflict of interest statement in the “Confidential to Editor” section, and submit your "Accept" recommendation.

Reviewer #2: All comments have been addressed

Reviewer #3: (No Response)

2. Is the manuscript technically sound, and do the data support the conclusions?

Reviewer #2: Partly

Reviewer #3: Yes

3. Has the statistical analysis been performed appropriately and rigorously? 

Reviewer #2: I Don't Know

Reviewer #3: Yes

4. Have the authors made all data underlying the findings in their manuscript fully available?

Reviewer #2: Yes

Reviewer #3: Yes

5. Is the manuscript presented in an intelligible fashion and written in standard English?

Reviewer #2: Yes

Reviewer #3: Yes

6. Review Comments to the Author

Reviewer #2: Thank you For your responsibilityو,this manuscript is now acceptable for publication but you should have highlighted the changes.

Reviewer #3: I would like to begin by thanking the authors for this useful and timely study. The paper shows detailed information about how Community Isolation Facilities (CIFs) were managed in Nakhon Si Thammarat, Thailand, during the COVID-19 pandemic. This topic is important, especially for improving public health systems in the future. The study gives real examples from the field, which can help health officers and policymakers learn what worked and what did not.

However, to make the paper clearer and more helpful to international readers, I would like to suggest the following improvements. One key point is that the authors should make the main findings of the study more clear. It would also be helpful to explain the reasons for success or problems in CIF operations. Also, showing how the Thai situation compares to other countries would help readers learn more from the study.

1. Introduction

- The paper does not explain clearly why Nakhon Si Thammarat was chosen as the study area. Please give more information, such as how many cases were there, how important is this province, or whether it is a good example of CIF management in Thailand.

- The authors should also explain the general situation of CIF and home isolation (HI) in Thailand. This will help readers outside the country understand the background.

2. Scoring Criteria (Lines 133–134 and 145–146)

- The authors divide the scores into “poor,” “moderate,” and “good” levels. Please explain what method or theory supports this way of scoring. Was it based on Bloom’s taxonomy or another system?

3. Statistical Analysis (Lines 150–152)

The authors say they used Bloom’s cut-off points but did not give a reference. Please provide a source or explanation for this choice.

4. Discussion – Key Findings

The main results of the study are not clearly written in the discussion. The authors mostly repeat the numbers from the survey without interpreting their meaning. It would be better to talk about the lessons learned or the underlying causes of the findings. Giving practical ideas for solving these problems in the future will make this paper stronger. For example,

- Lines 286–293: This paragraph talks about national policy, so it would be better in the Introduction, not in the Discussion.

- Lines 306–307: The authors mention problems in environmental health management but do not explain what they are. Please give more details and examples.

- Line 325–326: The authors state that overall preparedness for CIFs was good, while preparedness for location and infrastructure was only moderate. Why was this the case? Please explain the reasons. It would be helpful to include related factors, supporting evidence, or theoretical explanations that help readers understand this point more clearly.

- Lines 326–327: The authors mention that staff preparedness was poor in some CIFs (38.6%), which is quite surprising. Why was the preparedness level low, even though most staff had bachelor’s or master’s degrees in health sciences? Please give more details to explain this issue. For example, were there problems with training, experience, job clarity, or workload? This information would help readers better understand the challenges in CIF operations.

- Lines 350–352: The authors report that 28.1% of CIFs did not have a wastewater treatment system and 66.7% lacked a disinfecting system. This is a serious issue, as proper treatment is very important to prevent virus spreading. Please explain why many CIFs did not have these systems. Were there limitations in resources, planning, or technical support? Also, please suggest how this problem can be solved or prepared for in the future.

- Lines 410–412: The study shows concern about food provided by outsourced services or donations, mainly because delivery time could not be controlled and some food types could not be restricted. However, the discussion does not explain clearly why this issue was so important. Was it based on the experience of staff during the CIF operation? Were there past problems with food safety or nutrition? Please provide more explanation, as this finding could be very useful for improving food management in future public health emergencies.

- Lines 449–461: I am not fully sure about the main purpose of this paragraph, and I am not certain if I understood it correctly. The content seems quite general. For example, it explains that when the number of COVID-19 cases increased, the number of CIFs also increased. This seems to be a normal situation and does not clearly show the effectiveness of CIFs. In addition, the paragraph states that Figure 2 shows the relationship between CIF operations and their effectiveness, but it does not provide enough evidence to support this point. Please explain what is meant by "effectiveness" here. For example, did CIFs in Nakhon Si Thammarat help reduce the number of new cases or control the outbreak faster? If so, please provide data or analysis to support this claim. Otherwise, this section may appear more descriptive than analytical.

5. Figure

- Please check the quality of the figure again to ensure it meets the journal’s technical standards. The image should be clear, with readable text and proper resolution. Make sure the format, font size, and labeling follow the PLOS ONE figure guidelines. This will help improve the overall presentation of the manuscript.

7. PLOS authors have the option to publish the peer review history of their article (what does this mean?). If published, this will include your full peer review and any attached files.

Reviewer #2: **Yes: **hMaryam Javadi (PhD ),

1 Professor( Ph.D. in Nutrition), Children Growth Research Center, Research Institute for Prevention of Non-Communicable Diseases, Qazvin University of Medical Sciences, Qazvin, Iran

Qazvin, Iran .Iran.Zip Code: 34197-59811

2 Professor ( Ph.D. in Nutrition) Department of Nutrition, School of Health, Qazvin University of Medical Sciences, Qazvin, Iran.Zip Code: 34159-14595

Dr.MaryamJavadi,

professor , Department of Nutrition,School of Health, Qazvin University of Medical Sciences,

Qazvin, Iran. Qazvin.BolvarShahidBahonar.Qazvin University of Medical Sciences, Qazvin, Iran Zip Code: 34159-14595

Reviewer #3: No

---

## [Author Response · Author response to Decision Letter 2]

13 Sep 2025

Comment from Reviewer #2

Thank you for your responsibility, this manuscript is now acceptable for publication but you should have highlighted the changes.

Response: Thank you. We have highlighted the changes in the file “Revised Manuscript with Track Changes”

Comments from Reviewer #3

1. Introduction

Comment 1.1: The paper does not explain clearly why Nakhon Si Thammarat was chosen as the study area. Please give more information, such as how many cases were there, how important is this province, or whether it is a good example of CIF management in Thailand.

Response: Thank you for pointing this out. We have added the following information in the Introduction

Line 108 – 115 (in the revised manuscript): “In October 2021, during the study's preparatory phase, the Ministry of Public Health designated 10 of Thailand's 77 provinces as a "watchlist," which included Tak, Ratchaburi, Chantaburi, Rayong, Nakhon Ratchasima, Nakhon Si Thammarat, Narathiwat, Pattani, Yala, and Songkhla [22]. Local Government Organizations (LGOs) were central to the establishment of CIFs. Nakhon Si Thammarat was purposively selected as the study site. Although it had the second-largest number of LGOs on the watchlist, surpassed by Nakhon Ratchasima [23], it was chosen for its geographical convenience to the researcher, which facilitated data collection amidst the pandemic's travel restrictions.”

The new citations were added in the reference list as shown below.

22. World Health Organization. Coronavirus disease 2019 (COVID-19) Situation Report – 207, Thailand. WHO Thailand; 2021 Oct 28 [Cited 12 September 2025]. Available from: https://cdn.who.int/media/docs/default-source/searo/thailand/2021_10_28_eng-sitrep-207-covid19.pdf?sfvrsn=ce673f5b_5.

23. Department of Local Administration. Local Administration Department, Thailand [Cited 12 September 2025]. Available from: https://www.dla.go.th/work/abt/index.jsp.

Comment 1.2: The authors should also explain the general situation of CIF and home isolation (HI) in Thailand. This will help readers outside the country understand the background.

Response: Thank you for this suggestion. We have added the following information to a sentence in the Introduction to emphasize that Thailand applied two isolation strategies, i.e., HI and CI.

Line 65 (in the revised manuscript): Thailand applied “both” home-based isolation (HI) and community-based isolation (CI) for the COVID-19 confirmed cases in the Green Group.

In addition, we have included more information on the application of the two isolation strategies in Thailand as follows.

Line 80 – 84 (in the revised manuscript): “In Thailand, CI was the primary isolation strategy during the Delta variant outbreak. CIFs were gradually phased out due to the decline in the situation and budget constraints. Then, HI became the primary option with the emergence of the Omicron variant, which led to rapid transmission and a surge in cases that were asymptomatic or had mild symptoms.”

2. Method - Scoring Criteria (Line 133-134 and 145-146 in the original manuscript)

Comment: The authors divide the scores into “poor,” “moderate,” and “good” levels. Please explain what method or theory supports this way of scoring. Was it based on Bloom’s taxonomy or another system?

Response: It was based on Bloom’s taxonomy. The information (Line 133-134 and 145-146 in the original manuscript) has been removed and the criteria-based rating scale based on Bloom’s cut-off categories are presented in the Statistical Analysis (highlighted in yellow), shown as follows.

Line 203 – 205 (in the revised manuscript): “The scores were interpreted to assess preparedness and implementation levels using a criteria-based rating scale based on Bloom’s cut-off categories: poor (less than 60%), moderate (60 – 79%), and good (80 – 100%) [26, 27].”

3. Statistical Analysis (Line 150-152 in the original manuscript)

Comment: The authors say they used Bloom’s cut-off points but did not give a reference. Please provide a source or explanation for this choice.

Response: Two references have been included, shown as follows.

Line 203 – 205 (in the revised manuscript): “The scores were interpreted to assess preparedness and implementation levels using a criteria-based rating scale based on Bloom’s cut-off categories: poor (less than 60%), moderate (60 – 79%), and good (80 – 100%) [26, 27].”

The two references have been included in the Reference section (number 26 and 27).

26. Akalu Y, Ayelign B, Molla MD. Knowledge, Attitude and Practice Towards COVID-19 Among Chronic Disease Patients at Addis Zemen Hospital, Northwest Ethiopia. Infect Drug Resist. 2020;13: 1949–1960. https://doi.org/10.2147/IDR.S258736.

27. Jeenmuang K, Kaewsawas S, Thanapop C, Thanapop S. Social Support, Active Ageing Perception and Practices among Educational Staff in the Primary Educational Service, Nakhon-Si-Thammarat Province, Southern Thailand. Soc Sci. 2023;12(9): 486. https://doi.org/10.3390/socsci12090486.

4. Discussion – Key findings

The main results of the study are not clearly written in the discussion. The authors mostly repeat the numbers from the survey without interpreting their meaning. It would be better to talk about the lessons learned or the underlying causes of the findings. Giving practical ideas for solving these problems in the future will make this paper stronger.

Comment 4.1:

Lines 286–293 (in the original manuscript): This paragraph talks about national policy, so it would be better in the Introduction, not in the Discussion.

Response: The paragraph has been moved to the Introduction in the first revision (highlighted in yellow in the manuscript).

Line 85 – 92 (in the revised manuscript): “Thailand is the only upper-middle-income country in the top five of the Global Health Security (GHS) Index among 195 countries in 2021 [16]. This high ranking reflects the effectiveness of the government’s public health policy and emergency management during the COVID-19 pandemic. Prompt response to patients with mild or asymptomatic symptoms of COVID-19 by adopting the CI strategy played a significant role in effective disease control in Thailand. The containment of COVID-19 patients in community isolation facilities (CIFs) could suppress the spread of SARS-CoV-2 to the community by preventing transmission via direct contact with the patients and indirect contact through environmental pathways.”

Comment 4.2:

Lines 306–307 (in the original manuscript): The authors mention problems in environmental health management but do not explain what they are. Please give more details and examples.

Response: Thank you for pointing this out. The 'shortcomings in environmental health management' mentioned by the reviewer introduced the study's findings, which were discussed in the subsequent section; thus, specific details were not provided. To avoid reader confusion, this part has therefore been removed.

Comment 4.3:

Line 325–326 (in the original manuscript): The authors state that overall preparedness for CIFs was good, while preparedness for location and infrastructure was only moderate. Why was this the case? Please explain the reasons. It would be helpful to include related factors, supporting evidence, or theoretical explanations that help readers understand this point more clearly.

Response: Thank you for this suggestion. We have added the following information in the Discussion.

Line 379 – 393 (in the revised manuscript): “Data from Table 3 indicates that the absence of a wastewater treatment system—particularly one for disinfection prior to discharge—substantially lowered the overall preparedness scores for Location and Infrastructure to a moderate level. This situation may be because the local CIFs in this study were established within existing community infrastructure, as previously mentioned, to ensure a swift response in accommodating patients during an urgent situation. The priority was to isolate infected individuals from their families and the community quickly. However, these local facilities often lacked adequate spatial readiness as they were not initially designed for patient habitation. Therefore, to prevent the spread of disease in similar future outbreaks, relevant agencies such as LGOs should prepare suitable sites or improve existing infrastructure. This process should involve a comprehensive assessment of their suitability, covering environmental health aspects, to ensure a timely and effective response to potential future pandemics.”

To prevent redundancy with the information above on the pre-existing infrastructure, the following sentence(s) has/have been deleted:

Since the infrastructures were not built for patient wards, utilities for accommodation, such as food service and environmental sanitation, basic health care resources, and waste management systems, need to be provided.

Comment 4.4:

Lines 326–327 (in the original manuscript): The authors mention that staff preparedness was poor in some CIFs (38.6%), which is quite surprising. Why was the preparedness level low, even though most staff had bachelor’s or master’s degrees in health sciences? Please give more details to explain this issue. For example, were there problems with training, experience, job clarity, or workload? This information would help readers better understand the challenges in CIF operations.

Response: Thank you for pointing this out. The sentence “Interestingly, the level of preparedness in terms of staff was poor in some CIFs (38.6%)” was moved to the following two paragraphs, close to the discussion on the staff.

Line 428 (in the revised manuscript): “Interestingly, the level of preparedness in terms of staff was poor in some CIFs (38.6%).”

To address the reviewer's comment on staff preparedness, we have incorporated the following clarification:

Line 428 – 446 (in the revised manuscript): “This finding may be because, according to Table 3, the assessment of staff preparedness by founding staff was based on two criteria: 1) the sufficiency of staff and volunteers for CIF operations, and 2) the presence of qualified operating staff with an understanding of environmental health management. Regarding the first criterion on personnel numbers, during the CIFs' establishment phase, some LGOs, particularly smaller ones, had limited staff and had not yet established collaborations with other agencies to supplement their workforce. As for the operating staff, some founding staff (31.6%) reported that they were not qualified. However, self-reported educational backgrounds by the operating staff show that a majority (90.7%) held degrees in health sciences. A closer look at individual data revealed that only one person had a degree in environmental health. Furthermore, Table 2 indicates that the majority (61.4%) had no prior experience in environmental health. This data suggests a shortage of personnel specializing in environmental health within LGOs. It is essential to recruit more of these specialists for environmental health operations, especially if LGOs are to continue their role in establishing CIFs for future pandemic response. These measures would ensure the availability of qualified personnel capable of managing the environment to control infectious disease outbreaks effectively.”

Comment 4.5:

Lines 350–352 (in the original manuscript): The authors report that 28.1% of CIFs did not have a wastewater treatment system and 66.7% lacked a disinfecting system. This is a serious issue, as proper treatment is very important to prevent virus spreading. Please explain why many CIFs did not have these systems. Were there limitations in resources, planning, or technical support? Also, please suggest how this problem can be solved or prepared for in the future.

Response: Thank you for this suggestion. Information related to this point has already been added to the manuscript in response to comment 4.3.

Comment 4.6:

Lines 410–412 (in the original manuscript): The study shows concern about food provided by outsourced services or donations, mainly because delivery time could not be controlled, and some food types could not be restricted. However, the discussion does not explain clearly why this issue was so important. Was it based on the experience of staff during the CIF operation? Were there past problems with food safety or nutrition? Please provide more explanation, as this finding could be very useful for improving food management in future public health emergencies.

Response: Thank you for pointing this out. Proper food safety and nutrition are essential for protecting patients from complications arising from foodborne hazards. Since food sanitation is a critical element of environmental health, the core subject of this study, we included its management within CIFs in our investigation. It is important to note, however, that no cases of COVID-19 patients being directly harmed by the food served in the CIFs were reported. Accordingly, to better align our discussion with these findings, we have revised a sentence “This study showed that these issues were particularly concerning for food from outsources or supplied by donation since delivery time could not be strictly controlled and certain types of food could not be prohibited” as shown below:

Line 519 – 521 (in the revised manuscript): “This study found challenges in ensuring food safety for food from outside suppliers or donations, as delivery times and food types could not be strictly controlled. Despite these risks, no related foodborne illness cases were reported.”

Comment 4.7:

Lines 449–461(in the original manuscript): I am not fully sure about the main purpose of this paragraph, and I am not certain if I understood it correctly. The content seems quite general. For example, it explains that when the number of COVID-19 cases increased, the number of CIFs also increased. This seems to be a normal situation and does not clearly show the effectiveness of CIFs.

Response: This paragraph aims to explain the information in Figure 2: how CIFs could potentially delay reaching the case peak, a point further discussed in the next section of the manuscript as follows.

Line 610 – 615 (highlighted in yellow in the revised manuscript): “…..the operation of CIFs appeared to delay the peak of Delta cases. The Delta strain took approximately six months from April to October 2021, to reach its peak. In contrast, the Omicron period, without CIFs, took only two months, from January to the peak in March 2022. Compared to the Omicron outbreak, fewer people were vaccinated during the Delta period. It’s suggested that without CIFs in response to the Delta outbreak, the number of confirmed cases could have risen even in the early stages.”

In addition, the paragraph states that Figure 2 shows the relationship between CIF operations and their effectiveness, but it does not provide enough evidence to support this point. Please explain what is meant by "effectiveness" here. For example, did CIFs in Nakhon Si Thammarat help reduce the number of new cases or control the outbreak faster? If so, please provide data or analysis to support this claim. Otherwise, this section may appear more descriptive than analytical.

Response: We concede that the current figure does not sufficiently illustrate the effectiveness of CIFs. A definitive demonstration of their effectiveness in reducing case numbers or accelerating outbreak control would require a comparative dataset from other provinces and countries that did not implement CIFs during the Delta strain outbreak. We attempted to acquire such data, but no directly comparable information was available to estimate this effect. Consequently, we have decided to remove the sentence discussing the relationship between the measures and their effectiveness from our manuscript to prevent any over-interpretation of the existing data. We have revised the sentence “Fig 2 illustrates the relationship between the measures and their effectiveness by showing the number of CIFs versus the number of confirmed cases in Nakhon Si Thammarat Province during the 2021-2022 pandemic.” as follows:

Line 577 – 579 (in the revised manuscript): “Fig 2 illustrates the number of confirmed cases in Na

---

## [Editor Report · Decision Letter 2]

16 Sep 2025

Environmental health management in local community isolation facilities during COVID-19 pandemic: A case study in Nakhon Si Thammarat, Thailand

PONE-D-25-02239R2

Dear Dr. Vattanasit,

We’re pleased to inform you that your manuscript has been judged scientifically suitable for publication and will be formally accepted for publication once it meets all outstanding technical requirements.

Kind regards,

Phuping Sucharitakul

Academic Editor

PLOS ONE
---

## [Editor Report · Acceptance letter]

PONE-D-25-02239R2

PLOS ONE

Dear Dr. Vattanasit,

I'm pleased to inform you that your manuscript has been deemed suitable for publication in PLOS ONE. Congratulations! Your manuscript is now being handed over to our production team.

Kind regards,

on behalf of

Dr. Phuping Sucharitakul

Academic Editor

PLOS ONE